# Star-Shaped Poly(2-ethyl-2-oxazoline) and Poly(2-isopropyl-2-oxazoline) with Central Thiacalix[4]Arene Fragments: Reduction and Stabilization of Silver Nanoparticles

**DOI:** 10.3390/polym11122006

**Published:** 2019-12-04

**Authors:** Alexey Lezov, Alexander Gubarev, Maria Mikhailova, Alexandra Lezova, Nina Mikusheva, Vladimir Kalganov, Marina Dudkina, Andrey Ten’kovtsev, Tatyana Nekrasova, Larisa Andreeva, Natalia Saprykina, Ruslan Smyslov, Yulia Gorshkova, Dmitriy Romanov, Stephanie Höppener, Igor Perevyazko, Nikolay Tsvetkov

**Affiliations:** 1Department of Molecular Biophysics and Polymer Physics, St. Petersburg State University, Universitetskaya emb., 7/9, 199034 St. Petersburg, Russiaa.gubarev@spbu.ru (A.G.); m.e.mikhailova@spbu.ru (M.M.); a.lezova@spbu.ru (A.L.); nina.mikusheva@yandex.ru (N.M.); vdkalganov@yandex.ru (V.K.); i.perevyazko@spbu.ru (I.P.); 2Institute of Macromolecular Compounds of the Russian Academy of Sciences, Bolshoi pr. 31, 199004 St. Petersburg, Russia; 15lab@hq.macro.ru (M.D.); tenkovtsev@yandex.ru (A.T.); polar@imc.macro.ru (T.N.); andreeva@imc.macro.ru (L.A.); elmic@hq.macro.ru (N.S.); urs1968@gmail.com (R.S.); 3Institute of Biomedical Systems and Technologies, Peter the Great St. Petersburg Polytechnic University, Polytechnicheskaya 29, 195251 St. Petersburg, Russia; 4Joint Institute for Nuclear Research, Joliot-Curie 6, 141980 Dubna, Moscow Region, Russia; 5Institute of Silicate Chemistry of the Russian Academy of Sciences, Adm. Makarova emb. 2, 199034 St. Petersburg, Russia; dprom@mail.ru; 6Laboratory of Organic and Macromolecular Chemistry (IOMC), Friedrich Schiller University Jena, Humboldt Straße 10, 07743 Jena, Germany; s.hoeppener@uni-jena.de; 7Jena Center for Soft Matter (JCSM), Philosophenweg 7, 07743 Jena, Germany

**Keywords:** silver nanoparticles, star-shaped polymer, colloidal solution, temperature-responsive polymers

## Abstract

The interaction of silver nitrate with star-shaped poly(2-ethyl-2-oxazoline) and poly(2-isopropyl-2-oxazoline) containing central thiacalix[4]arene cores, which proceeds under visible light in aqueous solutions at ambient temperature, was studied. It was found that this process led to the formation of stable colloidal solutions of silver nanoparticles. The kinetics of the formation of the nanoparticles was investigated by the observation of a time-dependent increase in the intensity of the plasmon resonance peak that is related to the nanoparticles and appears in the range of 400 to 700 nm. According to the data of electron and X-ray spectroscopy, scanning and transmission electron microscopy, X-ray diffraction analysis, and dynamic light scattering, the radius of the obtained silver nanoparticles is equal to 30 nm. In addition, the flow birefringence experiments showed that solutions of nanoparticles have high optical shear coefficients.

## 1. Introduction

Biocompatible polymeric materials are actively investigated and applied in medicine and pharmaceutics. Polymers that are able to respond reversibly to minor changes in the properties of the environment (such as temperature, pH, ionic strength, and the presence of certain substances in solution) are attracting increasing attention of researchers. One important application of temperature-responsive polymers is the development of drug delivery systems that can release drugs in a controlled manner under the action of temperatures close to the physiological temperature of the human body. Poly(2-oxazoline)s are temperature-responsive and biocompatible polymers, which can be used as a versatile tool for designing polymer objects with complex architecture and adjustable physico-chemical properties [1,2,3,4]. Poly(2-oxazoline)s can be considered as substitutes for poly(ethylene glycol) (PEG) in medicine [5,6]. They have already found applications in the synthesis of hydrogels [7]; these systems may also be used as implants.

The broad prospects for the practical application of temperature-responsive polymers of complex architecture stimulated research works related to the synthesis and studies of the properties of star-shaped poly(2-oxazoline)s with hydrophobic cores [8,9,10,11]. It is known that branched and star-shaped polymers are more effective complexing agents in comparison with linear polymers. These compounds can be used in designing delivery systems for hydrophobic drugs that are capable of selective content release under the action of temperature [12,13,14].

As for the synthetic procedures for branched and star-shaped polymers, functionalized calix[n]arene macrocycles can be used as initiators of these reactions [14]. Calix[n]arenes have proven to be effective complexing agents for transition metal ions and low molecular weight organic compounds. In addition, they are non-toxic and can be used in vivo for medical purposes. However, the extremely low water solubility of simple calixarenes prevents their use in biomedical applications. The addition of amphiphilic polymeric “arms” to calixarene macrocycles may solve the problem of their high hydrophobicity and, at the same time, help to realize the beneficial properties of both classes of compounds.

Temperature-responsive star-shaped poly(2-ethyl-2-oxazoline) (star-PETOX) and poly(2-isopropyl-2-oxazoline) (star-PIPOX) with polymeric arms grafted to the lower rim of thiacalix[4]arene have been recently investigated in water and tetrahydrofuran solutions [15]. It was established that in tetrahydrofuran, the studied polymers are present only in the form of individual macromolecules, while in water, in addition to isolated molecules, large aggregates were found. Two structural levels of the molecular organization of star-PETOX and star-PIPOX in aqueous solutions were discovered. They are (i) the level of individual macromolecules and (ii) the level of supramolecular associates (clusters) with star-like architecture.

Due to the presence of sulfur atoms in star-shaped poly(2-ethyl-2-oxazoline) and poly(2-isopropyl-2-oxazoline) with arms grafted onto the lower rim of thiacalix[4]arene macromolecules, these polymers are able to reduce metal ions (in particular, silver ions) and stabilize the resulting metal nanoparticles [16].

Silver nanoparticles (Ag NPs) have attracted considerable attention due to their peculiar optical, chemical, electrical, and catalytic properties that can be attuned by changing surface nature, size, shape, and other characteristics of particles. Hence, nanosilver has been used in various fields, such as catalysis, design of sensors, electronic components, and the development of antimicrobial agents for the health industry [17,18,19]. Disinfectants based on silver NPs have received much attention because they find numerous practical applications in our daily life. Beyond this, Ag NPs have been used in different areas of industry, such as the development of silver-based air/water filters, textiles, in animal husbandry, biomedical industry, and manufacture of food packaging. If silver NPs are not properly stabilized, they undergo rapid oxidation and easily form aggregates in solutions, which hinders their application. The development of the methods for obtaining and effective stabilization of nanoparticles with narrow size distributions is an important task [19,20,21]. Medical application of preparations and materials based on silver NPs is only possible if these preparations were manufactured without the use of toxic reducing agents (ammonia solutions, alkali, etc.).

In recent years, the stabilization of Ag NPs by polymers has also been widely used. Polymers not only stabilize nanoparticle dispersions but also exert an influence on their sizes and shapes; thus, nanoparticles can acquire different physical properties depending on the intended application [22,23]. Polyethylene glycol (PEG), polyvinyl alcohol (PVA), and poly(*N*-vinylpyrrolidone) (PVP) are commonly used as stabilizers in the synthesis of Ag NPs. PVP is preferred over the others due to the presence of highly polar amide-containing rings in its side chains; these fragments have a high affinity for silver ions and silver nanoparticles. The mechanism of formation and stabilization of Ag NPs in the presence of PVP has previously been reported [23]. However, the stability of nanosilver dispersions containing poly(*N*-vinyl-2-pyrrolidone-*co*-vinyl acetate) and poly(2-ethyl-2-oxazoline) (PETOX) is significantly higher than the stability of dispersions stabilized by PVP. PETOX is a non-toxic and biocompatible polymer exhibiting lower critical solution temperature behavior with a critical temperature at 60 °C [24,25,26,27]. PETOX also contains amide groups linking the backbone and side chains, which are expected to contribute to better stabilization of Ag NPs [28]. In most papers, the complexes of silver nanoparticles with poly(2-oxazoline)s are received by the reduction of silver by low molecular weight agents [28,29]. In our investigation, the star-shaped molecule itself acts as a reduction agent.

The present paper is aimed to investigate the interaction between AgNO_3_ and star-PETOX (star- PIPOX) in aqueous solution. It was shown that at room temperature and under irradiation with visible light, silver ions are reduced with the forming of silver nanoparticles. The compositions were obtained for the first time and studied by absorption spectrophotometry, scanning and transmittance electron microscopy (SEM), X-ray diffraction analysis (XRD), dynamic light scattering (DLS), small-angle neutron scattering (SANS), and flow birefringence (FB, or the Maxwell effect).

## 2. Materials and Methods

### 2.1. Materials

Star-shaped poly(2-ethyl-2-oxazoline) (star-PETOX) and poly(2-isopropyl-2-oxazoline) (star-PIPOX) (see Scheme 1) with arms grafted to the lower rim of thiacalix[4]arene were synthesized according to the procedure reported previously [15]. Star polymers contain four arms, the average degree of polymerization of PETOX chains in the star-PETOX macromolecules was 16 monomer and for PIPOX in star-PIPOX macromolecule was 7 monomer units per 1 arm.

Silver nanoparticles were synthesized by mixing aqueous solutions of star-PETOX (or star-PIPOX) with aqueous solutions of AgNO_3_ at continuous stirring) (see Appendix A). Six mixtures were prepared; the silver/sulfur (*n*_Ag_/*n*_S_) numeric ratios (the ratio of the numbers of Argentum to Sulfur atoms in the solution) were varied over a wide range. For star-PETOX-based complexes, they were: *n*_Ag_/*n*_S_ = 0.11, 1.1, 11; for star-PIPOX-based complexes, *n*_Ag_/*n*_S_ = 0.11, 1.1, 11. The final concentrations of a star-shaped polymer in all mixtures were kept constant (*c*_polymer_ = 0.919 ± 0.014 g/dL). Thus, it was possible to compare the obtained results and to analyze the influence of silver content and chemical composition of the arms attached to thiacalix[4]arene on various processes (reduction, formation, and stabilization of silver nanoparticles in the presence of star-PETOX and star-PIPOX). After mixing, the solutions were irradiated by monochrome laser with the wavelength λ = 445 nm in the dark for 24 h (until the constant value of *Abs*_max_ in the absorption spectrum was achieved).

The formation of silver nanoparticles is a result of interaction between aqueous solutions of silver nitrate and star polymers occurring due to the photochemical reduction of Ag^+^ by visible light. The process is photosensitized by sulfur atoms. As seen in Figure 1a, irradiation of the star-PIPOX/AgNO_3_ mixture leads to the appearance of the surface plasmon resonance (SPR) peak related to Ag^0^ nanoparticles (λ_max_ = 412 nm) already after 2 h. At the same time, when the process was carried out in the dark, optical absorption in this region remained virtually nonexistent even after 7 h into the experiment. Similar behavior was observed for star-PETOX/AgNO_3_ mixtures.

The asymmetric shape of the SPR band and its considerable width are noteworthy (Figure 1b). The asymmetry of the absorption band in the long-wavelength region with respect to the maximum can be caused by a number of reasons. They include high polydispersity of metal nanoparticles, their asymmetry, or large sizes. In addition, stabilized nanoparticles may form aggregates, which also leads to broadening of the SPR spectrum in the long-wavelength region. Absorption spectra of the complexes based on star-PETOX and star-PIPOX obtained at the *n*_Ag_/*n*_S_ ratios equal to 0.1 and 1.1 were similar. Meanwhile, an increase in silver content for both systems manifested itself in the growth of intensity of the long-wavelength component of the spectrum (see Appendix A in Appendix A). It should be noted that we observed a positive correlation between the silver percentage in the reaction mixture and the magnitude of absorption at wavelengths exceeding 400 nm. The above is true for the star-PIPOX-based systems at *n*_Ag_/*n*_S_ = 1.1 and 11 only when absorption is registered at wavelengths exceeding 450 nm. It can be expected that the concentration of nanoparticles in the system increases, and certain redistribution of sizes between the components of the system occurs. A more accurate and detailed description of these processes can be obtained by the hydrodynamic methods.

When the thiacalix[4]arene core of the polymer is substituted for its carbocyclic analog that does not contain sulfur atoms (calix[4]arene), the reduction of silver nanoparticles does not take place under similar conditions. The ability of sulfur derivatives to sensitize photochemical reduction of silver in aqueous solutions is well known [30]. It is expected that during the interaction between the polymer containing thiacalix[4]arene core and silver nitrate in aqueous solution, clusters of various structures are formed. These clusters are similar to the products of the interaction between low molar mass derivatives of thiacalix[4]arene and Ag^+^ [31]. Upon photochemical reduction of the metal, the clusters decompose to metal silver and the initial polymer (that stabilizes nanoparticles at the expense of metallophilic Ag–S interactions). The studied polymers with a thiacalix[4]arene core serve simultaneously as reducing agents for silver ions and stabilizers for the formed nanoparticles.

### 2.2. Methods

Analytical Ultracentrifugation (AUC). Sedimentation velocity experiments were performed using a Beckman XLI analytical ultracentrifuge (ProteomeLab XLI Protein Characterization System, Brea, CA, USA). The rotor speed was 3000 to 60,000 rpm depending on the sample; temperature was 15 °C; interference (λ_i_ = 660 nm) and absorbance (λ_a_ = 430 nm) optical systems were engaged; double-sector cells with aluminum centerpieces (optical path length: 12 mm) were used. The studied solution and a solvent (0.42 mL) were loaded into the sample and reference sectors, respectively. The centrifuge chamber with loaded rotor and interferometer was vacuumized and thermo-stabilized for at least 60 to 90 min before the run. The average time of sedimentation process in experiments with the studied samples was about 17 to 19 h; the experiment was performed until a complete translation of a material from the meniscus to the cell bottom occurred. Concentration profiles within the cell were registered at 2 to 3 min intervals.

The sedimentation velocity data were analyzed using the Sedfit program [32]. Sedfit allows obtaining the distribution of sedimentation coefficients using the Provencher regularization procedure [33,34]. The main two characteristics of sedimentation velocity experiments were determined, viz. the sedimentation coefficient *s* and the frictional ratio *f*/*f*_sph_. In some cases, it is possible to reliably estimate diffusion coefficients with the use of frictional ratio values, but self-consistency of the acquired data should be established in the experiments involving linear poly(2-ethyl-2-oxazoline) samples [35]. Both characteristics (*s* and *f*/*f*_sph_) should be extrapolated to zero solute concentration. Since the hydrodynamic investigations are usually performed in extremely dilute solutions, the linear approximations $s^{-1}=s_{0}^{-1}(1+k_{s}c+\ldots )$ and $\left(f/f_{sph})={\left(f/f_{sph}\right)}_{0}(1+k_{f}c+\ldots \right)$ (where $k_{s}$ is the Gralen coefficient, c is the solution concentration, f is the translational friction coefficient, $f_{sph}$ is the translational friction coefficient of an equivalent sphere) can be used for the extrapolations. In this way, it is possible to determine hydrodynamic parameters $s_{0}$ and ${\left(f/f_{sph}\right)}_{0}$ that characterize a macromolecule at the infinite dilution limit. As mentioned before, in some cases, the *D* value may be obtained from the frictional ratio calculated by Sedfit program: $D_{0}=\frac{k_{B}T{\left(1-\bar{\upsilon }\rho_{0}\right)}^{1/2}}{\eta_{0}^{3/2}9\pi \sqrt{2}{\left({\left(f/f_{sph}\right)}_{0}\right)}^{3/2}{\left(s_{0}\bar{\upsilon }\right)}^{1/2}}$ (where $k_{B}$ is the Boltzmann constant, *T* is the absolute temperature, $\bar{\upsilon }$ is the partial specific volume, $\rho_{0}$, $\eta_{0}$ are the solvent density and viscosity, respectively) [32].

To eliminate the common solvent properties, the intrinsic values of the velocity sedimentation coefficient [*s*] and the translational diffusion coefficient [*D*] may be used: $\left[s]\equiv s_{0}\eta_{0}/(1-\bar{\upsilon }\rho_{0}\right)$ and $[D]\equiv D_{0}\eta_{0}/T$.

Thus, molecular masses may be determined using the Svedberg Equation:
(1)$$M_{sD}=\frac{s_{0}}{D_{0}}\frac{kTN_{A}}{\left(1-\bar{\upsilon }\rho_{0}\right)}=R\frac{\left[s\right]}{\left[D\right]}$$
where $\left(1-\bar{\upsilon }\rho_{0}\right)$ is the buoyancy factor, *N*_A_ is the Avogadro’s number, and *R* is the universal gas constant. All experiments were carried out at 15 °C.

Densitometry. The density measurements were carried out in water and tetrahydrofuran (THF) solutions using a DMA 5000 M density meter (Anton Paar GmbH, Graz, Austria) according to the procedure developed by Kratky et al. [36]. The corresponding dependence of $\Delta \rho =\rho -\rho_{0}$ on polymer concentration c was plotted, and the value of the partial specific volume was calculated as $\Delta \rho /\Delta c=(1-\upsilon \rho_{0}$), which constituted $\bar{\upsilon }=\left(0.86\pm 0.01\right)\;{\mathrm{cm}}^{3}g^{-1}$.

Dynamic Light Scattering (DLS). DLS experiments were carried out using a “PhotoCor Complex” apparatus (Photocor Instruments Inc., Moscow, Russia). The device included a digital correlator (288 channels, 10 ns), a standard goniometer (10°–150°), and a thermostat with temperature stabilization of 0.05 °C. The single-mode linear polarized laser (λ_0_ = 654 nm) was used as an excitation source; the experiments were carried out at scattering angles (θ) ranging from 30° to 130°. Autocorrelation functions of scattered light intensity were processed using the inverse Laplace transform regularization procedure incorporated in DynaLS software (provided by Photocor Instruments Inc., Moscow, Russia) (which provides distributions of scattered light intensities by relaxation times *τ* (*ρ*(*τ*)). The dependence of 1/*τ* (where *τ* is the position of a maximum in the *ρ*(*τ*) distribution) on the scattering vector squared $q^{2}={\left(4\pi n/\lambda \sin(\theta /2)\right)}^{2}$ for all studied samples was a straight line passing through the origin, this indicating the diffusional character of the observed processes ($1/\tau =Dq^{2}$) [37,38,39]. Hydrodynamic radius *R*_h_ was calculated using the Stokes—Einstein Equation [40]:
(2)$$R_{h}=\frac{kT}{6\pi \eta_{0}D}$$
where *k_B_* is the Boltzmann constant, *T* is the absolute temperature, and *η*_0_ is the solvent viscosity.

Small-Angle Neutron Scattering (SANS). SANS measurements were performed using a YuMO time-of-flight spectrometer (IBR-2 pulsed reactor, Dubna, Moscow region, Russia).

The standard data acquisition time when using YuMO was 30 min per sample. Two ring wire He^3^-detectors [41] located at distances of 4 and 13 m from the sample position were used in our experiment. The scattered intensity (differential cross-section per sample volume) was registered as a function of the momentum transfer modulus Q = (4π/λ) sin(θ/2), where θ is the scattering angle, and λ is the incident neutron wavelength. The incident neutron beam distribution provides the available wavelength range of 0.5 to 8 Å, which corresponds to the momentum transfer range of 0.07 to 5 nm^−1^ (*q*-range). The raw data treatment was performed using the SAS program (Joint Institute for Nuclear Research Dubna, Moscow region, Russia, company, city, country) [42]. The experimentally obtained SANS spectra were converted to the absolute scale by normalization to the incoherent scattering cross-section of the standard vanadium sample. The measured spectra were additionally corrected considering the scattering from the setup and empty cells as well as the background scattering. The final SANS curves were presented on the absolute scale, and the background noise was subtracted [43]. Analysis of the SANS spectra was performed with the help of the models proposed in the following sections using SasView software (Cary, NC, USA) [44].

Absorption spectrophotometry. Absorption spectrophotometry was used to detect the presence of NPs in solutions and to monitor dynamics of their formation (an UV-1800 spectrophotometer, Shimadzu, Japan). Absorption spectra (Abs) were measured in the wavelength (λ) range from 190 to 1100 nm with a step of 1 nm. Quartz cells with a light path of 0.5 cm were used.

Scanning electron microscopy (SEM). The images of star-PETOX- and star-PIPOX-based complexes with silver nanoparticles were obtained using a Zeiss Merlin scanning electron microscope (Carl Zeiss SMT, Oberkochen, Germany)) operating at an accelerating voltage of 10 kV; the pressure in the chamber was 50–70 Pa. The electron beam current was equal to 262 pA. The samples were prepared by placing a droplet of solution onto a silicon substrate followed by drying at 45 °C for 12 h. The composition of nanoparticles was studied by energy-dispersive X-ray spectroscopy (EDX).

TEM investigations were performed on an FEI Tecnai G^2^20 Transmission Electron Microscope (Hillsboro, OR, USA) with a LaB6 electron source operated at an acceleration voltage of 200 kV. A total of 15 µL of the sample solution was blotted onto carbon coated TEM grids (Quantifoil, Großlöbichau, Germany).

X-ray diffraction studies (XRD). XRD experiments were carried out on the diffractometer DRON-3M (SPA “Burevestnik”) in a mode of reflection (geometry Bragg -Brentano) using Cu Kα radiation (average wavelength λ = 1.54183 Å, Ni β-filter). The parameters of the generator: anode voltage 38 kV, tube current of 18 mA. Used cracks on the divergent primary beam—2 × 8 mm^2^, for reflected—0.25 mm, Soller slit with a divergence of about 2.5 at the primary and reflected beams. The survey was conducted in quartz cuvettes on a glass substrate without rotational averaging. Registration was carried out in continuous mode in the range of angles 2θ = 5° to 50° with an angular velocity detector 1°/min.

Flow birefringence (FB, Maxwell effect). The technique of flow birefringence measurements is described in detail in [40]. The measurements were carried out at a wavelength (λ) of 550 nm. The relative path difference (Δλ/λ) of the elliptical rotary compensator was equal to 0.032. The experiments were carried out in the concentration (*c*) range from 0.5 × 10^−2^ to 2 × 10^−2^ g/cm^3^. Since the solutions containing Ag NPs are intensely colored, the measurements of these solutions were conducted at significantly lower concentrations ((0.1 − 0.3) × 10^−2^ g/cm^3^). Use of these low concentrations (when *c*[η] < 1) allowed us to take the value of (Δ*n*/Δτ)*_p_* = (Δ*n*_p_ − Δ*n*_0_)/[*g*(η − η_0_)] as the optical shear coefficient of a dissolved polymer (Δ*n/*Δτ). Here, Δ*n*_p_ and Δ*n*_0_ are the observed birefringence values for a solution and a solvent, respectively; Δ*τ* = *g*(η − η_0_) is the excess shear stress, *g* is the flow velocity gradient, η and η_0_ are the viscosities of a solution and a solvent, respectively. The viscosity of the investigated samples was measured with the use of a Lovis 2000 M rolling-ball microviscometer (Anton Paar, Graz, Austria).

## 3. Results

### 3.1. Complexes of Star-PETOX and Star-PIPOX with Silver Nanoparticles in the Solid State

The SEM studies involved the star-PETOX and star-PIPOX-based samples obtained at *n*_Ag_/*n*_S_ = 0.1. In both cases, considerable amounts of spherical nanoparticles were observed in the samples deposited on the silica surface. According to the EDX data, the nanoparticles consist of silver atoms (Ag^0^) (see Appendix A). The distributions of equivalent disc radii *R_circ_* for two samples are presented in Figure 2. The average *R_circ_* for the star-PETOX/Ag^0^ complex was 37 nm, and for the star-PIPOX/Ag^0^ complexes, this parameter was equal to 31 nm (see Figure 2a).

Figure 2b presents the image of a silver particle obtained in the presence of star-PETOX. The EDX data (see Appendix A) suggest that the spherical formation in the center corresponds to metal silver, and the rim is formed by a polymeric shell. The high contrast between the polymeric shell and the substrate is apparently caused by the fact that this shell was ionized during irradiation of the sample with the electron beam. Thus, it can be assumed that the distributions presented in Figure 2a characterize the sizes of the polymer/nanoparticles complex. High resolution TEM images display the crystalline structure of the obtained nanoparticles (Figure 2c,d) (see Appendix A).

XRD diffractograms of the star-PIPOX powder-like samples (see illustrations in Appendix A) contain an amorphous halo in the angular (2θ) range ~(10° to 50°). The amorphous halo in the diffractogram of the star-PIPOX/Ag^0^ composite (Appendix A) is considerably broader. The maximums at 2θ ~38.2° (111) and ~44.4° (200) are clearly seen; these peaks are typical of silver nanoparticles with a face-centered cubic lattice [45]. The half-width of the observed crystalline reflections indicates small sizes of the formed silver particles. The average size of particles was calculated using the Scherrer formula:
$$d=\frac{K\lambda }{\beta cos\theta }$$
where *d* is the average size of crystals; *K* is the dimensionless coefficient characterizing particle shape (the Scherrer constant); λ is the wavelength of X-ray radiation; β is the full width at half maximum (FWHM) of a reflection (rad, 2θ units); θ is the diffraction angle (the Bragg angle). It was found that in the solutions that were left to stay for a week, *d* was equal to 26 nm, while in the solutions stored for eight months, this parameter increased up to 28 nm. The broadening of amorphous halo indicates a reduction in the degree of structural order of the initial polymer. This phenomenon may be caused by the incorporation of silver nanoparticles between copolymer chains that stabilize these nanoparticles.

To summarize, there is a good correlation between the data obtained by different methods that were used to investigate the properties of the synthesized complexes in a condensed state. According to the SEM data, the polymeric complexes of silver nanoparticles based on star-PIPOX have lower dimensions than those based on star-PETOX.

### 3.2. Hydrodynamic Characteristics of Star-PETOX and Star-PIPOX Complexes with Silver Nanoparticles in Solutions

As was established earlier [15], star-PETOX and star-PIPOX macromolecules demonstrate multilevel organization in water solutions. Distributions of hydrodynamic radii obtained by DLS for star-PETOX solutions indicate the presence of particles with hydrodynamic radii of about 5 nm and 25 nm. In addition to large aggregates, the aqueous solution of star-PIPOX contains individual macromolecules in which the hydrodynamic radii *R_h_* are equal to about 2.3 nm. The results obtained by analytical ultracentrifugation showed that the individual star-PETOX and star-PIPOX macromolecules predominate in aqueous solutions (~90%).

The distributions of hydrodynamic radii observed in solutions of star-PETOX complexes with silver nanoparticles include two modes: (i) 6 to 8 nm for *n*_Ag_/*n*_S_ ratio equal to 0.1, and (ii) 40 to 60 nm for *n*_Ag_/*n*_S_ ratio equal to 1.1 (see Table 1, Figure 3a). The indices 1, 2, 3 in Table 1 correspond to the free polymer and to the components of the complex, correspondingly. An increase in the *n*_Ag_/*n*_S_ ratio up to 11 leads to the appearance of large species; their hydrodynamic radius is 110 nm.

Similar results were obtained for solutions of star-PIPOX/Ag^0^ complexes; double-peak distributions were observed in the *n*_Ag_/*n*_S_ range from 0.1 to 1.1 (see Table 1, Figure 3b). When the used *n*_Ag_/*n*_S_ ratio was equal to 11, the hydrodynamic radii distribution included three peaks.

The complex shape of the distribution functions indicates high dispersity of the studied systems. One simple method for processing DLS data obtained in the studies of disperse systems is the cumulant analysis [38]. This method makes it possible to estimate the apparent average hydrodynamic radius *R*_hCUM_ of the studied system and its polydispersity index PDI. Table 2 presents the characteristics calculated by this method for solutions of star-PETOX/Ag^0^ and star-PIPOX/Ag^0^ complexes prepared at various *n*_Ag_/*n*_S_ ratios.

Analysis of the DLS data by the cumulant method shows that both systems (based on star-PETOX and star-PIPOX) demonstrate the increase in PDI with increasing the *n*_Ag_/*n*_S_ ratio. This result correlates well with the absorption spectrophotometry spectroscopy data (namely, with the growth in absorption intensity in the long-wavelength region of a spectrum with increasing the *n*_Ag_/*n*_S_ ratio (see Appendix A). Complexes with the lowest dispersity were obtained at the ratio *n*_Ag_/*n*_S_ = 0.1 (for both copolymers). The *R*_hCUM_ value of the star-PETOX-based complexes varied only slightly with the composition of the reaction mixture. At the same time, an increase in the PDI value was observed. This result may indicate that an increase in the silver fraction in the initial mixture leads to a rather uniform broadening of particle size distribution for a solution of this complex. In the case of star-PIPOX-based complexes, we observed an insignificant decrease in the *R*_hCUM_ value with increasing the *n*_Ag_/*n*_S_ ratio. Apparently, the rise in silver content in the reaction mixture led to an increase in the portion of small particles in solution.

Analytical ultracentrifugation (AUC) was used in the study of the sedimentation velocity of solutions of star-PETOX/Ag^0^ and star-PIPOX/Ag^0^ complexes that were prepared at *n*_Ag_/*n*_S_ = 0.1; as in this case, the observed polydispersity of the complexes (PDI) is minimal. The experiments were carried out in water (H_2_O) and deuterated water (D_2_O) according to the “differential sedimentation” approach [46,47].

In the case of the complexes prepared on the basis of star-PETOX, the presence of two types of particles with different sizes was reliably established. The first type (NP1) sedimented at a rotation speed of 3000 rpm during the first 60 min of the experiment (see Figure 4a). Meanwhile, the optical density of the initial solution registered at λ = 430 nm had dropped by ~80%. Particles of the second type (NP2) sedimented during the next 8 h; the optical density of the solution further decreased further (by ~10% with respect to the initial value). Taking into account the measured residual optical density and edge effects of NP2 distribution, we can expect the presence of the third type of particles (NP3) in the sedimentation coefficient range from 10 to 100 S. However, it was not possible to resolve the NP3 distribution at the maximum available concentration of the solution.

According to the sedimentation velocity data, the complexes based on star-PIPOX showing lower dispersity in comparison to the star-PETOX-based systems. Optical density of the solution (the main peak) constitutes about ~90 to 95 wt %; the edge effects of distributions are pronounced weakly. The values of sedimentation coefficients are almost similar to the coefficients that characterize the size of predominant nanoparticles (NP1) in the star-PETOX-based system (see Figure 4b).

In order to calculate the molar masses of the obtained complexes using Equation (1), it is necessary to determine the values of their partial specific volume $\bar{\upsilon }$. To determine $\bar{\upsilon }$, we studied the sedimentation velocity in H_2_O (*ρ*_0_ = 0.9991 g/cm^3^, η_0_ = 1.142 cP) and H_2_O/D_2_O mixtures. For the experiments with star-PETOX-based systems, the volume ratio of components was 0.75%D_2_O/0.25%H_2_O, and in the case of the star-PIPOX/Ag^0^ complex, the ratio was 0.67%D_2_O/0.33%H_2_O. This volume ratio was varied because of different initial concentrations of the studied solutions and the lower threshold sensitivity of the AUC optical absorption system (0.2 to 0.25 OD).

The density and dynamic viscosity of the mixtures were calculated based on the additivity of these parameters and the data reported in [48]. For the first mixture (0.75% D_2_O/0.25% H_2_O), the following values of these parameters were determined: *ρ*_0m1_ = 1.0791 g/cm^3^, η_0m1_ = 1.3612 cP; for the second mixture (0.67%D_2_O/0.33%H_2_O), we obtained *ρ*_0m2_ = 1.0706 g/cm^3^, and η_0m2_ = 1.3378 cP. The determination of the partial specific volume is difficult due to the extremely high polydispersity of the studied complexes. Thereby, the partial specific volume was estimated from the sedimentation coefficients corresponding to the distribution maximums found in H_2_O and the 0.75%D_2_O/0.25%H_2_O mixture; the weight-average values of the main peaks found in H_2_O and the 0.67%D_2_O/0.33%H_2_O mixture were also used. In this manner, the values of partial specific volumes of nanoparticles (NP1) in the star-PETOX- and star-PIPOX-based systems can be estimated only with much uncertainty. For star-PETOX/Ag^0^, we obtained $\bar{\upsilon }$ = (0.2 ± 0.1) cm^3^/g, and for star-PIPOX/Ag^0^, the value was $\bar{\upsilon }$ = (0.3 ± 0.1) cm^3^/g. The obtained $\bar{\upsilon }$ values are coincidentally within the experimental accuracy; therefore, in further calculations, we used their average value (0.25 ± 0.15) cm^3^/g. The $\bar{\upsilon }$ value determined in this manner lies in the range from 0.1 to 0.86. The lower limit of this range is comparable with the inverse density of silver, and the upper limit corresponds to the partial specific volume of star-PETOX and star-PIPOX macromolecules [15]. For further estimations, we assumed that the found value of the specific partial volume of NP1 particles does not differ from that for NP2 in the star-PETOX-based system. For the complexes based on star-PETOX, the masses of the observed structures were the following: *M*_sDNP1_ = 1.3 × 10^9^ g/mol, *M*_sDNP2_ = 8.6 × 10^6^ g/mol. For the star-PIPOX/Ag complexes, we obtained *M*_sDNP_ = 1.2 × 10^9^ g/mol.

After precipitation of nanoparticles at 3000 rpm, we observed sedimentation of the initial star-PETOX polymer at 60,000. The obtained distributions are presented in Appendix A; we also compared them with the distribution obtained in the sedimentation experiments involving solutions of the initial polymer. The parameter characterizing shift of interference fringes for the initial solutions was used to evaluate the concentration of free polymer that does not participate in the stabilization of silver nanoparticles; on average, this concentration corresponds to the range of ~80% to 90% (Appendix A). It should be noted that, according to the obtained data (Appendix A), dilution of the solutions did not lead to changes in shape and other characteristics of the distributions of the sedimentation coefficients. Therefore, we can assume that the polydispersity of the obtained complexes is caused mainly by polydispersity of silver nanoparticles and not by particle aggregation.

The complexes obtained based on both studied systems, as well as the initial star-shaped polymers, demonstrate the lower critical solution temperature (LCST). The DLS studies revealed that the cloud points of the initial polymers and their complexes with silver are approximately similar. It should be noted that after cooling, the solutions became transparent again; according to the DLS data, the distributions of the particles present in solutions by hydrodynamic radii obtained before and after heating were similar.

### 3.3. SANS Study of Complexes of Star-Shaped Polymers with Silver Nanoparticles

Small angle neutron scattering was used to study structural changes in the star-shaped polymers in the presence of Ag^0^ nanoparticles in aqueous medium at the nanoscale level. This was possible thanks to the contrast between the polymer and water, $I\left(Q\right)∼({\bar{\rho }}_{pol}-\rho_{D_{2}O}$), as follows from the values of scattering length density (SLD) for neutrons in the studied systems presented in Appendix A.

The SANS experimental data for the complexes of silver nanoparticles with star-PETOX and star-PIPOX macromolecules in D_2_O are presented in Figure 5a. At first glance, the shapes of curves are satisfactorily described by an empirical model, where multiple Guinier and Porod regions can be identified (Appendix A). This model (Appendix A) gives useful information about the shape and size of the complexes in the region of low Q ($QR_{g}<1$). Thus, the dimension variable (s) is equal to zero for all systems under study (Appendix A). This means that the spherical shape of the star polymer aggregates is retained in the presence of silver nanoparticles. In addition, the shape of the Kratky plot ($Q^{2}I{\left(Q\right)}_{coh}$ vs. Q, Figure 5b) is typical of star-like polymers with the number of arms *f* >> 1 [49,50]. Note that the incoherent background noise was subtracted as a slope from the $Q^{4}I\left(Q\right)$ vs. $Q^{4}$ representation in the region of high Q [51].

Another important parameter of a polymer-Ag^0^ complex is the radius of gyration ($R_{g}$); the radii of gyration were estimated in several ways. Using a generalized Guinier-Porod model, we determined that for the pure star-PETOX polymer, $R_{g}$ is equal to 7.9 ± 0.1 nm; in the case of star-PETOX/Ag^0^ complex, we obtained $R_{g}$ = 8.5 ± 0.2 nm (Appendix A). These values are in good agreement with the results obtained from the Guinier plot (Figure 5c). Linear approximation of the experimental data plotted on coordinates ln I(Q) vs. $Q^{2}$ (in the region where the Q values satisfy the condition $QR_{g}$ < 1.3) gives the slope value. This value is related to the radius of gyration as: *Slope* = $-R_{g}^{2}/3$. Bearing in mind the value R = (5/3)^1/2^$R_{g}$ for spherical objects, we can conclude that the presence of silver nanoparticles leads to an increase in the size of polymer complexes. For instance, in the case of the star-PETOX/Ag^0^ complex, the diameter increased from 20.7 to 23.3 nm. For star-PIPOX/Ag^0^ samples, the average aggregate size was 21.5 nm.

However, the generalized Guinier/Porod approximation does not describe the SANS data that include peaks and oscillations in power-law decays. For this reason, we applied the unified Guinier-Porod model developed by G. Beaucage [52,53].

Unified Guinier-Porod approximation. In our previous paper [15], it has been demonstrated that the use of two mutually complementary methods (DLS and SANS) allowed observation of cylindrical clusters at the first supramolecular level of individual star-PETOX and star-PIPOX macromolecules. In addition, it turned out that at the second level, their clusters assembled to form dense star-shaped structures (star-PIPOX, fractal dimension *D*_m2_ = 2.9) and less dense structures in the case of star-PETOX (fractal dimension *D*_m2_ = 2.6) [15]. We used the unified Guinier-Porod model, including two structural levels (Appendix A), to fit the SANS data over the whole range of momentum transfer values Q. The results of this analysis are presented in Figure 5a (solid curves) and summarized in Table 3.

Deviation from the power law $Q^{-n}$ in scattering intensity is observed in the range of low momentum transfer values (*Q* < 0.3 nm^−1^). This deviation is caused by the transition to Guinier mode, where the scattering is determined by the characteristic size of mass-fractal clusters *R*_c_ (in the case of fractal systems, this is the upper limit of self-similarity). These mass-fractal clusters scatter independently. We analyzed dispersion in the Guinier mode and found that the *R*_g_ for mass-fractal clusters in the case of star-PETOX polymer in D_2_O (system S1) is equal to 7.99 nm (Table 3). This estimation is based on the slope of the ln I(Q) vs. $Q^{2}$ curve. However, the resulting value is slightly lower than that reported in our earlier paper (11.0 ± 0.9 nm [15]). This discrepancy is due to the fact that in the present work, we deal with the experimental SANS data, while in [15], we reported the results of modeling in the range of the momentum transfer values characteristic of SANS and ultra-SANS. In our previous work, we introduced the third structural level responsible for scattering by agglomerates, and this assumption caused a slight difference in the fitting results.

When scattering inhomogeneities are represented in the form of spherical mass-fractal objects, their characteristic size *R*_c_ can be expressed as follows:
(3)$$R_{C}={\left[\left(D_{m}+2\right)/D_{m}\right]}^{1/2}\times R_{g}$$


This characteristic size was calculated to be 10.4 ± 0.5 nm; in this case, the fractal dimension *D_m_* is equal to the power index in the generalized Porod law, *n* = 2.84 (Table 3). Thus, the particles formed by star-PETOX with sizes of 21 nm are observed in D_2_O. For the star-PETOX/Ag^0^ complex in D_2_O, the obtained value of the gyration radius (9.9 nm) is 1.2 times higher than for the star-PIPOX/Ag^0^ sample (Table 3). The characteristic diameter of inhomogeneities in star-PETOX solution was calculated by taking into account Equation (3) and found to be 2*R_C_* = 26 nm. Thus, the volume of scattering particles present in the solution of star-PETOX/Ag^0^ complex in D_2_O is higher than that of the initial polymeric star-PETOX particles by (13.0/5.34)^3^ = 14.4 times. The volume of the star-PIPOX/Ag^0^ particle (which has a characteristic diameter of 2*R_C_* = 25 nm) coincides with that of the star-PETOX/Ag^0^ complex.

As seen in Figure 5a, in the region of intermediate *Q* values (from 0.3 to 0.9 nm^−1^ for star-PETOX/Ag^0^—star-PIPOX/Ag^0^ SANS curves), we observe almost similar power indices in the generalized Porod law (*n_2_* ≈ 2.8). This fact indicates neutron scattering by inhomogeneities with mass-fractal packing, their fractal dimension being *D_m_* = *n* ≈ 2.8. Spatial packing of the studied systems is denser than that of a Gaussian sphere of the same gyration radius by ${\left(\frac{2D_{m}}{D_{m}+2}\right)}^{3/2}=={\left(\frac{2\;2.8}{4.8}\right)}^{3/2}=1.3$ times (see Table 3 and Equation (3)) it should be noted, that *D*_m_ = 2 for Gaussian sphere.

High molecular dispersion, *Ð*, of the systems under investigation leads to the fact that it is difficult to interpret the dimensionless shape parameter ρ = *R_g_*/*R_H_* [54]. The latter indicates the architecture of the supramolecular structure. In our case, it is 9.86/19.7 = 0.50 for PETOX and 9.54/16.3 = 0.58 for PIPOX. The ϱ values obtained are considerably lower than *ρ* = (3/5)^½^ = 0.775 for solid, homogeneous spheres. Following the reasoning of [54], this effect can be interpreted by the presence in the supramolecular organization of a branched system of hanging chains of “microgels”, which leads to a much smoother decay in the density of segments to larger radii than the “hard sphere” with its clearly defined surface. A similar result was observed for microgels formed by polyvinyl acetate [55].

### 3.4. Flow Birefringence (Maxwell Effect)

Optical properties of the solutions containing macromolecules and particles whose dimensions significantly exceed those of solvent molecules are similar to the properties of a system that consists of colloidal particles and a solvent with refraction index *n*_s_.

The Magnitude of the ratio between intrinsic birefringence and intrinsic viscosity is defined by anisotropy of polarizability of a particle *γ*_1_ − *γ*_2_ according to the following relationship:
$$\left[n]/[\eta ]∼(\Delta n/\Delta \tau )=\frac{4\pi {\left(n_{s}{}^{2}+2\right)}^{2}}{45kTn_{s}}\times (\gamma_{1}-\gamma_{2})\times F(p\right)$$
where *F*(*p*) is the function of the relationship between axes of a particle (macromolecule) *p* [40]. Generally, the difference between main polarizabilities (*γ*_1_ − *γ*_2_) of a macroscopic particle is a sum of two addends:*γ*_1_ − *γ*_2_ = (*γ*_1_ − *γ*_2_)*_i_* + (*γ*_1_ − *γ*_2_)_*f*_
where (*γ*_1_ − *γ*_2_)*_i_* is the intrinsic anisotropy caused by ordered structure of a substance constituting a particle (orientational order of valence bonds, atomic groups, etc.), and (*γ*_1_ − *γ*_2_)_*f*_ is the optical anisotropy related to shape of non-spherical particles whose refractive index differs from that of a solvent.

It should be noted that optical shear coefficients (Δ*n*/Δτ) for the initial star-PETOX and star-PIPOX polymers cannot be reliably measured; their values are extremely low and can be taken equal to zero.

Figure 6 presents dependences of the birefringence Δ*n* on the shear stress Δ*τ* for the solutions of silver nanoparticles stabilized by star-PIPOX. Note that the observed FB is very high, which is typical for large colloidal particles. In addition, the obtained Δ*n*(Δ*τ*) dependences deviate from linearity (see Figure 6), which also indicates the appearance of large asymmetrical particles in the system. As was pointed in [56], the presence of asymmetrical species may be caused both by certain non-sphericity of metal NPs, and by the formation of NP associates (even in insignificant amounts).

Removal of a portion of large associates by preparative centrifugation (9000 rpm for 30 min) resulted in a decrease in the value of the initial slope of the Δ*n*(Δ*τ*) dependence from 750 to 450 × 10^−10^ cm × s^2^/g. This change is primarily caused by a decrease in NP concentration in the solution. Note that the general behavior of the Δ*n*(Δ*τ*) dependence remains the same after centrifugation since the observed effect is caused by the presence of Ag nanoparticles and their associates in the system.

## 4. Conclusions

Amphiphilic star-shaped polymers with thiacalix[4]arene cores were studied as polymeric stabilizers of silver nanoparticles. It was found that sulfur atoms in polymeric ligands efficiently sensitize photochemical reduction of silver nitrate in aqueous solution; the resulting colloidal solutions of metal nanoparticles are stable over long periods of time. It was established that the PDI of polymer-based systems increases with increasing the *n*_Ag_/*n*_S_ ratio in both cases (star-PETOX/Ag^0^ and star-PIPOX complexes). Complexes with the narrowest size distributions were obtained on the basis of both polymers at the *n*_Ag_/*n*_S_ ratio equal to 0.1. The complexes based on star-PIPOX have lower polydispersity than those based on star-PETOX. The results of AUC experiments show that solutions of complexes contain high amounts of free macromolecules; thus, only about 20% of polymer chains take part in the stabilization of silver in the star-PIPOX/Ag^0^ system. The obtained complexes, as well as the initial star-shaped polymers, demonstrate lower critical solution temperature (LCST); the LCST of the complexes is similar to those of the initial polymers. High flow birefringence was observed in solutions of silver nanoparticles stabilized by star-PIPOX; this result is caused primarily by the presence of silver nanoparticles (that have a certain shape asymmetry) or insignificant amounts of their aggregates. The results obtained by the SANS method correlated well with the molecular hydrodynamics and optics data and made it possible to analyze the structural organization of the studied star-PETOX/Ag^0^ and star-PIPOX/Ag^0^ composites in solutions.

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
