# Peer review of "Star-Shaped Poly(2-ethyl-2-oxazoline) and Poly(2-isopropyl-2-oxazoline) with Central Thiacalix[4]Arene Fragments: Reduction and Stabilization of Silver Nanoparticles"

_polymers, 2019, doi:10.3390/polym11122006_

Round 1
Reviewer 1 Report
This paper describes a certainly peculiar, very interesting polymer – nanoparticle complex, in which a star polymer containing a core (thiacalix[4]arene) and linear arms (PETOX or PIPOX) are used to reduce Ag ions to Ag nanoparticles (AgNPs), which are further stabilized by the polymers themselves. For the characterization of the complexes, the authors employ analytical ultracentrigation (AUC), dynamic light scattering (DLS) SEM and TEM microscopies, diffraction, flow birefringence, etc, etc – a really impressive suite of techniques. Thus, one most valuable aspect of this paper is the description of their application to polymer-nanoparticle problems. The contents of this paper is perfectly suitable for Polymers and hopefully will attrack a wide audience. I strongly recommend publication. I just have a few comments about things that seem to be missing or unclear.
Although the authors provide a detailed list of literature references on oxazoline polymers -- including their own papers, refs 13 14 and 15, readers may like to find within this paper some elementary information, thus avoiding literature searches. For instances it is pertinent to show chemical schemes of oxazolyne star polymers (core and branches). Also basic information on number of branches, molecular weights, etc.
The term “silver/sulphur numeric” ratio is unclear. Although the meaning can be suspected, a well-stated definition is required. Something similar happens with the indices 1, 2, 3 for Rh, Rsf and MsD in Table 1. Do they correspond to the modes (i), (ii) and free polymer??
AUC measurements were only done with the 0.1 Ag/S ratio. Is there any non-trivial reason that could be worth reporting?.
The f/f_sph ratio is mentioned as being involved in AUC. It is well known that this ratio is informative about overall conformation of the particles. But I don’t find numerical results...
Among the variety of techniques, the inclusion small-angle neutron scattering (SANS) measurements is particularly remarkable – a sophisticated and not-easily-accesible instrumentation !! The main outcome from SANS is the radius of gyration. R_g. It is well known that a joint analysis of scattering and hydrodynamics, in the form of Rg/Rh ratio could provide some information. Is there some possibility in this regard?
Author Response
Dear Editors and Referees,
We are very grateful for attention to the manuscript and valuable comments regarding its content. We have thoroughly analyzed the raised questions and prepared step-by-step answers together with manuscript corrections (marked with yellow).
Q1
Although the authors provide a detailed list of literature references on oxazoline polymers -- including their own papers, refs 13 14 and 15, readers may like to find within this paper some elementary information, thus avoiding literature searches. For instances it is pertinent to show chemical schemes of oxazolyne star polymers (core and branches). Also basic information on number of branches, molecular weights, etc.
The chemical structures of studied star polymers as well as its characteristics was added in the manuscript
See page 3 Lines 113-117.
Q2
The term “silver/sulphur numeric” ratio is unclear. Although the meaning can be suspected, a well-stated definition is required. Something similar happens with the indices 1, 2, 3 for Rh, Rsf and MsD in Table 1. Do they correspond to the modes (i), (ii) and free polymer??
The definition of the term “silver/sulphur numeric” ratio was added in the manuscript.
Page 3, lines 120-121.
The indices 1, 2, 3 for Rh, Rsf and MsD in Table 1 correspond to free polymer and to the components of the complex, correspondingly. Corresponding explanations were included in the manuscript.
Page 9, lines 320-321.
Q3
AUC measurements were only done with the 0.1 Ag/S ratio. Is there any non-trivial reason that could be worth reporting?.
According to DLS and absorption spectroscopy data PDI increases with increasing the nAg/nS ratio. Complexes with the lowest polydispersity were obtained at the ratio nAg/nS = 0.1. AUC experiments were performed for the systems with minimal polydispersity. This slightly simplified the problem of determination of partial specific volume of studied system.
It should be also noted that value of optical density (at wave length 430 nm and optical path 1cm) for the studied systems obtained at nAg/nS=0.1 practically coincide with the maximal limit of the optical density in AUC absorption optical system. It allowed us to study star-PETOX and star-PIPOX systems without additional dilution.
Q4
The f/f_sph ratio is mentioned as being involved in AUC. It is well known that this ratio is informative about overall conformation of the particles. But I don’t find numerical results...
The (f/f_sph)_0 values have been added into Table SM-4-3(SM). The corresponding comments were introduced for Table 1: The detailed information on velocity sedimentation data is presented in Table SM-4-3(SM).
Q5
Among the variety of techniques, the inclusion small-angle neutron scattering (SANS) measurements is particularly remarkable – a sophisticated and not-easily-accesible instrumentation !! The main outcome from SANS is the radius of gyration. R_g. It is well known that a joint analysis of scattering and hydrodynamics, in the form of Rg/Rh ratio could provide some information. Is there some possibility in this regard?
The paragraph was added to the manuscript.
Page 15, lines 502-510.
Reviewer 2 Report
This manuscript described a work where it was studied for the first time the preparation of silver nanoparticles using two distinct poly(2-oxazoline) derivatives. The authors evaluate the respective kinetics of formation using UV-vis spectroscopy as the respective size, morphology, and crystalline phase using scanning and transmission electron microscopy, X-ray diffraction analysis, dynamic light scattering, and X-ray spectroscopy. Furthermore, they also evaluated the optical shear coefficients using flow birefringence.
This is an interesting and pioneering study where star-shaped polymers (PETOX and PIPOX) are used as stabilisers of silver nanoparticles using a photochemical reduction. The characterisation of the materials is well described, having the authors a very detailed discussion of this part.
In order to improve the document, other points in the text need to be changed and/or clarified.
As general comments; in practical all articles in the literature when it is referred to the size of the nanoparticles, it was referred to the diameter, not the radius. This form could give the wrong idea to the readers. Please include in the introduction a small paragraph about the work that is already done with poly(2-oxazoline)s and silver nanostructures to emphasize the difference to this one.
Abstract: The authors claim that the silver nanoparticles have 30 nm. However, looking to the TEM images the silver core was around 5 nm, they are the complexes that have around 30 nm of radius. Furthermore, the star-PETOX/Ag0 and star-PIPOX/Ag0 complexes and both have distinct sizes; please indicate both values.
Line 106: Please include the “(see SI-1)” in line 109. The SI-1 indicate the spectrums of star-PETOX and star-PIPOX with the Ag0, and in this case, after the preparation of silver nanoparticles.
Line 116: The solutions were irradiated by monochrome laser were also in the dark? If yes, please indicate.
Line 117: How many days are the solutions under laser irradiation? In Figure 1, it was presented around 500 min that is equivalent to around 8 h. Furthermore, the laser was able to be turned in for several days? If yes include the values of the spectra.
Line 117: As a curiosity, the incidence of the laser in the solutions during 8 h does not cause an increase in the temperature of the solutions? If yes, it can be a dual role in the reduction (visible light and temperature), and this needs to be indicated.
Figure 2: Please provide (if not in the main text put in SI) high-resolution TEM of the two nanoparticles, however, with less magnification in order to see more particles in the same image to understand the monodispersity of the samples.
Lines 266-268: Indicate the sizes of the core and the correspondent shell individually.
Line 273-274: If possible, put the diffractograms until 2q=80º in order to see the (220) and (311) phases. For the star-PIPOX, the authors could try to concentrate more the sample in the XRD support in order to observe the peaks.
Table 2: It is essential to the readers to understand that the values of the R with so high PDI (between 1 and 2) are meaningless. Usually, with such high values of PDI, the DLS quality report was no good.
Author Response
Dear Editors and Referees,
We are very grateful for attention to the manuscript and valuable comments regarding its content. We have thoroughly analyzed the raised questions and prepared step-by-step answers together with manuscript corrections (marked with yellow).
Q1
As general comments; in practical all articles in the literature when it is referred to the size of the nanoparticles, it was referred to the diameter, not the radius. This form could give the wrong idea to the readers. Please include in the introduction a small paragraph about the work that is already done with poly(2-oxazoline)s and silver nanostructures to emphasize the difference to this one.
The text of the introduction was modified.
Page 3, lines 99-102
Q2
Abstract: The authors claim that the silver nanoparticles have 30 nm. However, looking to the TEM images the silver core was around 5 nm, they are the complexes that have around 30 nm of radius. Furthermore, the star-PETOX/Ag0 and star-PIPOX/Ag0 complexes and both have distinct sizes; please indicate both values.
Distributions of equivalent disc radii (Rcirc) in polymer/Ag0 samples were obtained by SEM (Figure 2 (a, b)). Hundreds of particles were analyzed for each complex. The average Rcirc for star-PETOX/Ag0 complex was 37 nm, and for star-PIPOX/Ag0 complexes, this parameter was equal to 31 nm (see Fig. 2 (a)).High resolution TEM experiments were performed only for visualization of crystalline structure of nanoparticles. We are absolutely agree that presented in Figure 2 (c,d) images of small particles are in the left border of distributions Figure 2 (a), but it is the best pictures that we have.
Q3
Line 106: Please include the “(see SI-1)” in line 109. The SI-1 indicate the spectrums of star-PETOX and star-PIPOX with the Ag0, and in this case, after the preparation of silver nanoparticles.
Paragraph was corrected, chemical structures of star-PETOX and star-PIPOX were added.
Page 3, line119
Q4
Line 116: The solutions were irradiated by monochrome laser were also in the dark? If yes, please indicate.
Yes studied systems were irradiated in the dark, this clarification was added in the manuscript:
See page 3 lines 127-128.
Q5
Line 117: How many days are the solutions under laser irradiation? In Figure 1, it was presented around 500 min that is equivalent to around 8 h. Furthermore, the laser was able to be turned in for several days? If yes include the values of the spectra.
The solutions were irradiated by monochrome laser with the wavelength λ = 445 nm (at visible light) in the dark for 24 hours. The main criterion of the choice of irradiation time was setting of constant value of Absmax in the absorption spectrum. It should be mentioned, that in the case of the highest silver/sulfur numeric ratio nAg/nS=11 the constant value of Absmax in the absorption spectrum was achieved after ~8 hours. In figure 1 (a) ) Time dependence of optical density related to the optical density measured after 24 hours I_24 of irradiation obtained for the solution of star-PIPOX (nAg/nS=11) was presented. The values of the spectra I/I_24 was equal to unity after 500 min.
Correspondent part of the manuscript was modified.
Page 3, lines 127-128
Q6
Line 117: As a curiosity, the incidence of the laser in the solutions during 8 h does not cause an increase in the temperature of the solutions? If yes, it can be a dual role in the reduction (visible light and temperature), and this needs to be indicated.
The main idea of applying the laser as light source was to obtain spectrally narrow light and to put a light source away from the sample, it could prevent a direct heating from a light source. We used unfocused laser light, from 5mW solid state laser. The solutions of the complexes with volume about 3 ml were prepared with constant stirring at room temperature. We suppose that in mentioned above conditions the deviation of solution temperature from room temperature due to laser light absorption is insignificant.
Q7
Figure 2: Please provide (if not in the main text put in SI) high-resolution TEM of the two nanoparticles, however, with less magnification in order to see more particles in the same image to understand the monodispersity of the samples.
High resolution TEM images were added in the Figure SM-2-hrTEM.
In Figure SM-2-SEM SEM image and corresponding energy-dispersive X-ray spectra for complexes of silver nanoparticles with star-PETOX (a) and star-PIPOX (b) macromolecules were presented. It is clearly seen that studied nanoparticles are polydisperse, this fact is confirmed by other experimental techniques.
Q8
Lines 266-268: Indicate the sizes of the core and the correspondent shell individually.
For the particle which was presented in figure 2 (b) the requested parameters are presented on the picture. Similar data were observed for a number of particles on SEM images, however this data have only qualitive character. It should be also noted that this parameters will not be the same in solution.
Q9
Line 273-274: If possible, put the diffractograms until 2q=80º in order to see the (220) and (311) phases. For the star-PIPOX, the authors could try to concentrate more the sample in the XRD support in order to observe the peaks.
Unfortunately, we do not have such spectra.
Q10
Table 2: It is essential to the readers to understand that the values of the R with so high PDI (between 1 and 2) are meaningless. Usually, with such high values of PDI, the DLS quality report was no good.
Two approaches of DLS data handling were applied to studied systems: cumulant analysis and regularization procedure. Both of this approaches have a number of disadvantages with application to a highly dispersed systems. Regularization procedure usually gives a single wide peak or series of narrow ones for Rh distributions for disperse systems. Cumulant analysis provides quantitative result for Rh only for low disperse systems. As was appointed in the manuscript studied complexes are very dispersed and the main idea of DLS data fitting by mentioned above techniques was to observe our systems from the different points of view.
Cumulant analysis allowed us to obtain a qualitative evaluation of PDI, which made it possible to give a simple representation of the fact that increasing n_Ag/n_S relation leads to growth of the dispersity of the systems.